# Chronic 2P-STED imaging reveals high turnover of dendritic spines in the hippocampus in vivo

Thomas Pfeiffer[1,2†], Stefanie Poll[3†], Stephane Bancelin[1,2†], Julie Angibaud[1,2], VVG Krishna Inavalli[1,2], Kevin Keppler[4], Manuel Mittag[3], Martin Fuhrmann[3‡*], U Valentin Nägerl[1,2‡*]

[1]Interdisciplinary Institute for Neuroscience, CNRS UMR 5297, Bordeaux, France; [2]Interdisciplinary Institute for Neuroscience, University of Bordeaux, Bordeaux, France; [3]Neuroimmunology and Imaging Group, German Center for Neurodegenerative Diseases, Bonn, Germany; [4]Light Microscope Facility, German Center for Neurodegenerative Diseases, Bonn, Germany

*For correspondence:
martin.fuhrmann@dzne.de (MF);
valentin.nagerl@u-bordeaux.fr
(UVN)

[†]These authors contributed
equally to this work
[‡]These authors also contributed
equally to this work

Competing interests: The
authors declare that no
competing interests exist.

Reviewing editor: Karel
Svoboda, Janelia Research
Campus, Howard Hughes
Medical Institute, United States

**Abstract** Rewiring neural circuits by the formation and elimination of synapses is thought to be a key cellular mechanism of learning and memory in the mammalian brain. Dendritic spines are the postsynaptic structural component of excitatory synapses, and their experience-dependent plasticity has been extensively studied in mouse superficial cortex using two-photon microscopy in vivo. By contrast, very little is known about spine plasticity in the hippocampus, which is the archetypal memory center of the brain, mostly because it is difficult to visualize dendritic spines in this deeply embedded structure with sufficient spatial resolution. We developed chronic 2P-STED microscopy in mouse hippocampus, using a 'hippocampal window' based on resection of cortical tissue and a long working distance objective for optical access. We observed a two-fold higher spine density than previous studies and measured a spine turnover of ~40% within 4 days, which depended on spine size. We thus provide direct evidence for a high level of structural rewiring of synaptic circuits and new insights into the structure-dynamics relationship of hippocampal spines. Having established chronic super-resolution microscopy in the hippocampus in vivo, our study enables longitudinal and correlative analyses of nanoscale neuroanatomical structures with genetic, molecular and behavioral experiments.
DOI: https://doi.org/10.7554/eLife.34700.001

## Introduction

Dendritic spines form the postsynaptic structural component of most excitatory synapses in the mammalian brain. They constitute computational units of information processing that underlie essentially all higher brain functions (*Nimchinsky et al., 2002*; *Sala and Segal, 2014*; *Yuste and Bonhoeffer, 2004*) and play a crucial role in brain disorders such as autism spectrum disorder and Alzheimer's disease (*Dorostkar et al., 2015*; *Südhof, 2008*). Spine structure is closely linked to synapse function, as the size of spine heads scales with synaptic strength (*Matsuzaki et al., 2001*; *Noguchi et al., 2011*) and the shape and number of spines can be modified by the induction of synaptic plasticity (*Matsuzaki et al., 2001*; *Engert and Bonhoeffer, 1999*; *Nägerl et al., 2004*; *Tønnesen et al., 2014*; *Zhou et al., 2004*) and by sensory experience (*Holtmaat et al., 2006*; *Keck et al., 2011*).

Rewiring of neural circuits by spine plasticity is considered a key neurobiological mechanism of memory formation (reviewed in [*Nimchinsky et al., 2002*; *Sala and Segal, 2014*; *Yuste and Bonhoeffer, 2004*; *Kasai et al., 2010*]). Notably, a recent study showed that optically induced shrinkage

of potentiated spines disrupts newly acquired motor skills (*Hayashi-Takagi et al., 2015*), indicating a causal link between spine plasticity and memory. While experience-dependent plasticity of dendritic spines has been a consistent finding across mouse cortex in vivo (*Holtmaat and Svoboda, 2009*), very little is known about it in the hippocampus. However, this is a major knowledge gap, because this neural structure constitutes the archetypical memory center of the brain, and hippocampal brain slices and primary cell cultures are the dominant model systems for the study of synaptic plasticity mechanisms.

Imaging dendritic spines in the hippocampus in vivo is challenging because of its remote location more than 1 mm below the surface of the mouse brain. A pioneering study accomplished this with two-photon (2P) microscopy, but only over a period of a few hours (*Mizrahi et al., 2004*). Recently, approaches based on a 'hippocampal window' (*Gu et al., 2014*) or micro-endoscopy (*Attardo et al., 2015*) enabled 'chronic' 2P imaging over several weeks. However, 2P microscopy inevitably lacks the spatial resolution to visualize important details of spine morphology, such as spine necks (*Bethge et al., 2013*), and even struggles to resolve individual dendritic spines on CA1 pyramidal neurons, leaving a high fraction of them undetected (*Attardo et al., 2015*).

While regular light microscopy typically detects a spine density of around 1 spine/µm (*Gu et al., 2014*; *Attardo et al., 2015*; *Brigman et al., 2010*), electron microscopy (EM) reports ~3 spines/µm on CA1 hippocampal pyramidal neurons in rats (*Harris et al., 1992*) and in *stratum radiatum* of mice (*Bloss et al., 2018*). By comparison, spine density is about ten times lower in many cortical areas, for example ~0.24 spines/µm for pyramidal neurons in layer 5 of mouse barrel cortex (*Holtmaat et al., 2006*).

Given the limited spatial resolution of 2P microscopy, we turned to super-resolution stimulated emission depletion (STED) microscopy (*Klar et al., 2000*; *Hell, 2007*) to improve the visualization of dendritic spines in the intact hippocampus of living mice. We used a home-built STED microscope based on 2P excitation (2P-STED) (*Bethge et al., 2013*; *Ter Veer et al., 2017*) and equipped it with a long working distance objective to reach the deeply located hippocampus. We adopted a 'hippocampal window' technique (*Gu et al., 2014*; *Schmid et al., 2016*; *Dombeck et al., 2010*), where a portion of the overlying somatosensory cortex is surgically removed and replaced by a metal cylinder sealed with a cover slip, providing stable optical access to the CA1 area of the hippocampus.

We demonstrate that our new approach offers substantially improved spatial resolution and image quality compared to regular 2P microscopy in mouse hippocampus in vivo. Using transgenic mice with fluorescently labeled pyramidal neurons, we measured spine density on basal dendrites of pyramidal neurons in *stratum oriens* of the CA1 area and compared results obtained with 2P and 2P-STED microscopy in vivo as well as with STED microscopy in fixed hippocampal sections. Furthermore, we carried out repetitive 2P-STED in vivo imaging over a 4-day period to measure spine turnover.

Our analysis showed a two times higher spine density than reported by conventional 2P microscopy, and around 40% of all spines turned over within 4 days, suggesting a high level of circuit remodeling in the hippocampus in vivo. Furthermore, detailed morphological analysis revealed that primarily small spines were affected by spine turnover.

## Results

### 2P-STED microscopy with a long working distance objective

We set up in vivo STED microscopy of dendritic spines in mouse hippocampus to track their morphological dynamics over the course of several days. We used a custom-built 2P-STED microscope (*Figure 1A*) (*Bethge et al., 2013*; *Ter Veer et al., 2017*) in combination with a modified 'cranial window' technique to gain high-quality optical access to *stratum oriens* in the CA1 area of the hippocampus (*Gu et al., 2014*; *Dombeck et al., 2010*). We surgically removed the overlying somatosensory cortex and inserted a metal tube sealed with a cover slip as a physical place holder (*Figure 1B*) (*Gu et al., 2014*; *Dombeck et al., 2010*). To bridge the distance between the surface of the skull and the alveus located right above the hippocampus, we used an objective with a long working distance yet relatively high numerical aperture (Nikon N60X-NIR: WD 2.8 mm, NA 1.0).

To verify that this objective is compatible with STED imaging, we measured the point-spread function (PSF) of the microscope using fluorescent nanospheres (diameter: 40 nm). The full-width at

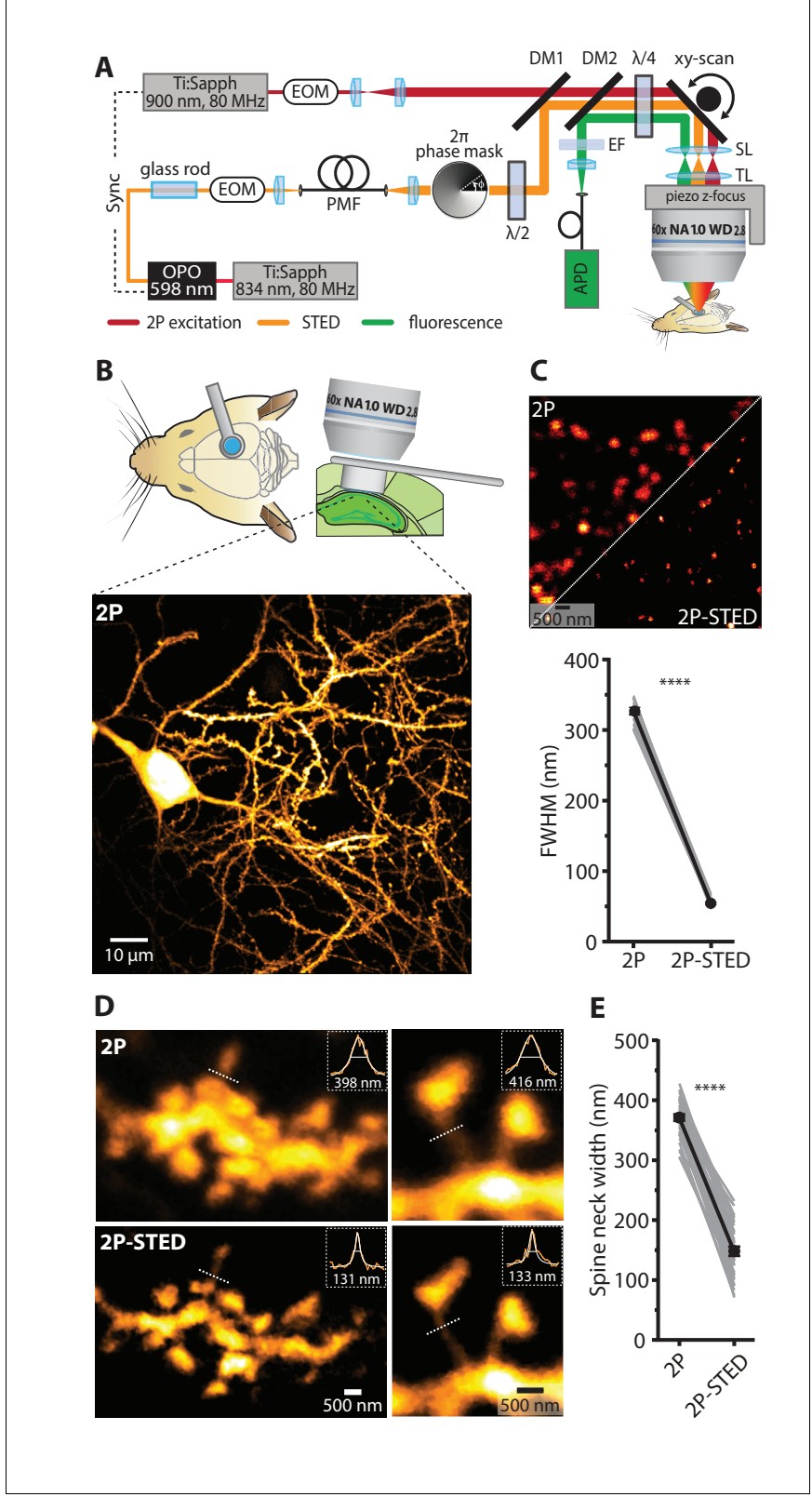

**Figure 1.** 2P- STED microscopy of dendritic spines in the hippocampus in vivo. (**A**) Schematic of the custom-built upright 2P-STED microscope. A Ti:Sapphire laser emits light pulses at 834 nm with 80 MHz repetition rate. The laser pumps an optical parametric oscillator (OPO) to obtain pulsed STED light at 598 nm. A glass rod and a polarization-maintaining fiber (PMF) stretch the STED pulses. The STED doughnut is engineered by a helical 2π

*Figure 1 continued on next page*

*Figure 1 continued*

phase mask in combination with λ/2 and λ/4 wave plates. The second Ti:Sapphire laser tuned to 900 nm with 80 MHz repetition rate served for two-photon (2P) excitation and is synchronized to the STED light pulses. The 2P and STED beam are combined using a dichroic mirror (DM1). Both beams are swept over the specimen using a galvo-based x-y scanner and a z-focusing device. Scan (SL) and tube lens (TL) image the scan mirrors into the back focal plane of the objective and ensure that the expanded laser beams overfill the back aperture of the objective. Fluorescence is de-scanned and guided to the avalanche photodiode (APD) via a dichroic mirror (DM2) and filters (EF). Electro-optical modulators (EOM) serve for quick adjustments of the beam intensity. (B) Schematic visualizing the combined use of the hippocampal window preparation and a long working distance objective (top). A 2P overview image showing the cell body and basal dendrites of a GFP-labeled pyramidal neuron in hippocampal CA1 in vivo (bottom). Maximum intensity projection (MIP) of ten z-sections with 2 μm z-steps. (C) Quantification of the lateral resolution with 40 nm fluorescent nanospheres. Upper panel: Representative comparison of 40 nm beads imaged by 2P and 2P-STED microscopy. Lower panel: Paired comparison of mean full-width at half maximum (FWHM) of line profiles obtained from 40 nm nanospheres (p<0.0001, paired t-test; n = 17 beads). (D) Representative images obtained from a GFP-labeled dendrite acquired by 2P and 2P-STED microscopy. Insets show line profiles fitted with Lorentzian functions and FWHM values obtained from the indicated dotted lines of the spine neck (MIP of three z-sections). (E) Paired measurements of spine neck widths imaged in 2P and 2P-STED mode (p<0.0001, paired t-test; n = 29 spine necks, 11 dendrites, 6 mice).

DOI: https://doi.org/10.7554/eLife.34700.002

The following video, source data, and figure supplements are available for figure 1:

**Source data 1.** Data for panel C.
DOI: https://doi.org/10.7554/eLife.34700.006
**Source data 2.** Data for panel B.
DOI: https://doi.org/10.7554/eLife.34700.007
**Source data 3.** Data for *Figure 1—figure supplement 1*.
DOI: https://doi.org/10.7554/eLife.34700.008
**Figure supplement 1.** Axial PSF measurement: 2P *versus* 2P-STED.
DOI: https://doi.org/10.7554/eLife.34700.003
**Figure supplement 2.** Measurements of spine neck diameters and direct comparison of 2P versus 2P-STED.
DOI: https://doi.org/10.7554/eLife.34700.004
**Figure 1—video 1.** Hippocampal window approach afforded high level of sample stability.
DOI: https://doi.org/10.7554/eLife.34700.005

half maximum (FWHM) was $54 \pm 2$ nm in the x-y plane in 2P-STED mode, compared with $325 \pm 5$ nm in regular 2P mode (p<0.0001, paired t-test; n = 17 beads; *Figure 1C*), demonstrating that our super-resolution approach delivered a six-fold nominal gain in lateral spatial resolution. In contrast, the axial resolution remained unchanged in 2P-STED (*Figure 1—figure supplement 1*).

## 2P-STED microscopy of dendritic spines in the hippocampus in vivo

We then used the 2P-STED microscope to image dendritic spines on basal dendrites of CA1 pyramidal neurons (*stratum oriens*) in anesthetized mice. We used transgenic mice (*Thy1*-H$^{YFP/+}$ and *Thy1*-M$^{GFP/+}$, 4–12 months old) that expressed either GFP or YFP in a subset of CA1 pyramidal neurons (*Feng et al., 2000*). Focusing on dendrites close to the coverslip (5–20 μm), spines were much more clearly delineated in 2P-STED than in 2P mode (*Figure 1D*, *Figure 1—figure supplement 2*). In particular, spine necks were better resolved with 2P-STED as evidenced by paired measurements of spine neck widths (2P-STED: $147 \pm 8$ nm; 2P: $369 \pm 6$ nm; p<0.0001, paired t-test; n = 35 spine necks, 17 dendrites, 6 mice; *Figure 1E*).

Image distortions caused by animal breathing and heartbeat normally pose a serious challenge for imaging in vivo, in particular STED microscopy, which is especially sensitive to brain motion due to its high spatial resolution. However, we only encountered mild levels of image blur (*Figure 1—video 1*), making it not necessary to apply motion correction (see Discussion).

## Dendritic spine density of CA1 pyramidal neurons in vivo

We determined spine density of basal dendrites of CA1 pyramidal neurons, comparing it in images acquired both in 2P-STED and 2P mode (*Figure 2A*). There was a significant difference between the two modes (2P-STED: $2.13 \pm 0.10$ μm$^{-1}$; 2P: $1.61 \pm 0.07$ μm$^{-1}$; p<0.0001, paired t-test; n = 82

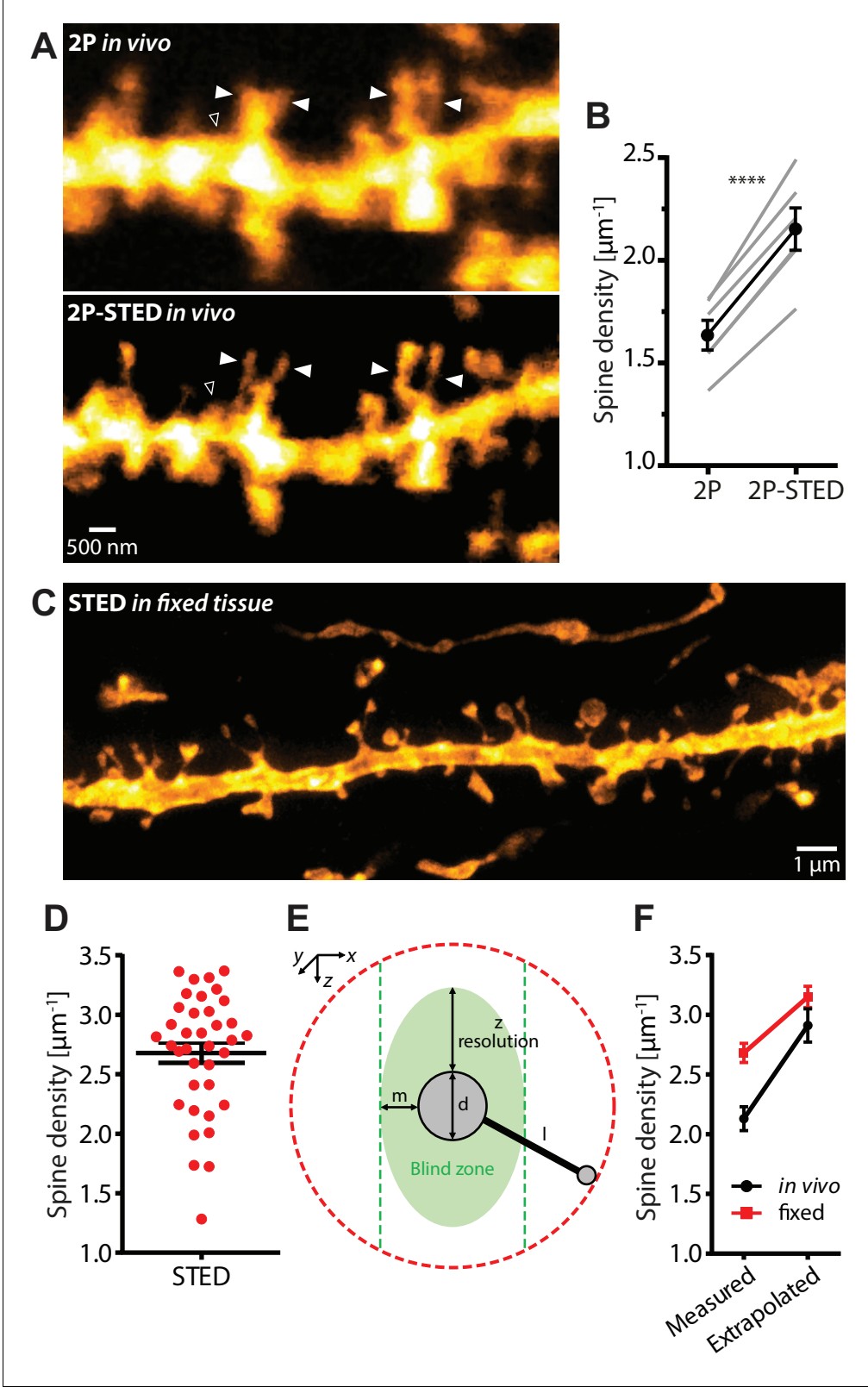

**Figure 2.** Density of spines on basal dendrites of CA1 pyramidal neurons in vivo. (**A**) CA1 basal dendrite imaged with 2P (top) or 2P-STED (bottom) microscopy. Filled arrowheads highlight spines discerned in 2P-STED, but not 2P mode. Open arrowheads indicate spines that could only be visualized in 2P-STED mode as they were otherwise masked by the blurry fluorescence of the dendrite in the 2P image (MIP of two z-sections). (**B**) Measured spine

*Figure 2 continued on next page*

*Figure 2 continued*

densities in consecutive 2P and 2P-STED acquisitions of the same dendrites (p<0.0001, paired t-test; n = 82 basal dendrites, 6 mice). (**C**) Image of a basal dendrite obtained from fixed brain tissue acquired on a confocal STED. (**D**) Dendritic spine density on basal dendrites in fixed hippocampal tissue (n = 37 basal dendrites, 6 mice). (**E**) Geometric model to extrapolate the spine density in 3D. Spines cannot be detected when they are inside the 'blind zone', depending on the dimensions of the morphology and microscope PSF. (**F**) Extrapolation of spine density in 3D following the model in (**E**) for in vivo (black line) and fixed tissue measurements (red line).

DOI: https://doi.org/10.7554/eLife.34700.009

The following video, source data, and figure supplement are available for figure 2:

**Source data 1.** Data for panel B.
DOI: https://doi.org/10.7554/eLife.34700.012
**Source data 2.** Data for panel D.
DOI: https://doi.org/10.7554/eLife.34700.013
**Figure supplement 1.** Methodology of dendritic spine density analysis in vivo.
DOI: https://doi.org/10.7554/eLife.34700.010
**Figure 2—video 1.** z-stack of a fixed dendrite imaged with one-photon STED microscopy.
DOI: https://doi.org/10.7554/eLife.34700.011

dendrites, 6 mice; *Figure 2B* and *Figure 2—figure supplement 1*), where 2P-STED detected on average 32% more spines than 2P in a direct comparison (*Figure 2B*). This increase was due to the improved detection or discrimination of (1) spines that either had short necks or that extended into the z-direction, which were otherwise obscured by the fluorescence signal of the dendritic shaft, and (2) closely clustered spines, which appeared merged in the 2P images (*Figure 2A*).

While our 2P-based mean value for spine density (1.61 $\mu m^{-1}$) is around 45% higher than the values reported in the light microscopy literature (~1.1 $\mu m^{-1}$) (*Gu et al., 2014*; *Attardo et al., 2015*), the mean value based on 2P-STED is about twice as large as this (2.13 $\mu m^{-1}$). We also counted spines in fixed brain slices obtained from the same animals (*Figure 2C* and *Figure 2—video 1*). Using a commercial STED microscope with one-photon excitation, we measured a density of 2.68 ± 0.08 $\mu m^{-1}$ (n = 37 dendrites, 6 mice; *Figure 2D*), which comes closer to the values reported by EM.

To understand where the remaining discrepancy might come from, we made a simple geometrical model to account for the limited axial resolution of our STED approach, which had left some spines invisible (*Figure 2E*). According to the model, which took into account the actual dimensions of dendritic morphology and the PSF of the microscope, about a quarter of the spines could not be detected in vivo because of this problem. Correcting the measured values by this fraction, we calculated a spine density of 2.91 ± 0.14 $\mu m^{-1}$ for the in vivo data and 3.15 ± 0.09 $\mu m^{-1}$ for the fixed tissue data (*Figure 2F*), which effectively closed the gap to the 'ground truth' EM values.

## Dendritic spines undergo high morphological turnover in vivo

To determine the kinetics of spine emergence and elimination, which is a matter of controversy (*Gu et al., 2014*; *Attardo et al., 2015*), we performed repeated 2P-STED imaging of individual dendritic stretches over 4 days (day 0, 2 and 4; *Figure 3A*). To retrieve a particular dendrite in consecutive imaging sessions, we used tissue landmarks such as blood vessels, fluorescent cell bodies and dendrites.

First, we quantified the average spine density for each time point, which remained stable over the entire observation period (day 0: 2.31 ± 0.10 $\mu m^{-1}$; day 2: 2.28 ± 0.08 $\mu m^{-1}$; day 4: 2.30 ± 0.10 $\mu m^{-1}$; n = 14 dendrites, 3 mice; *Figure 3A,B* and *Figure 3—source data 1*).

Next, we examined spine turnover by counting all spines that appeared and disappeared from one imaging session to the next. Furthermore, we calculated the 'survival fraction', which is the percentage of spines that were present on day 0 and visible during the subsequent imaging sessions. The survival fraction was 78.1 ± 3.6% for day 2, and 60.8 ± 3.4% for day 4 (*Figure 3C*).

We then calculated the fraction of spines lost and gained between the imaging sessions. The fraction of lost spines was 21.2 ± 3.6% from day 0 to 2, and 24.7 ± 3.0% from day 2 to 4 (*Figure 3D*), while the fraction of new spines from day 0 to 2 was 20.4 ± 2.6%, and 21.9 ± 3.0% from day 2 to 4

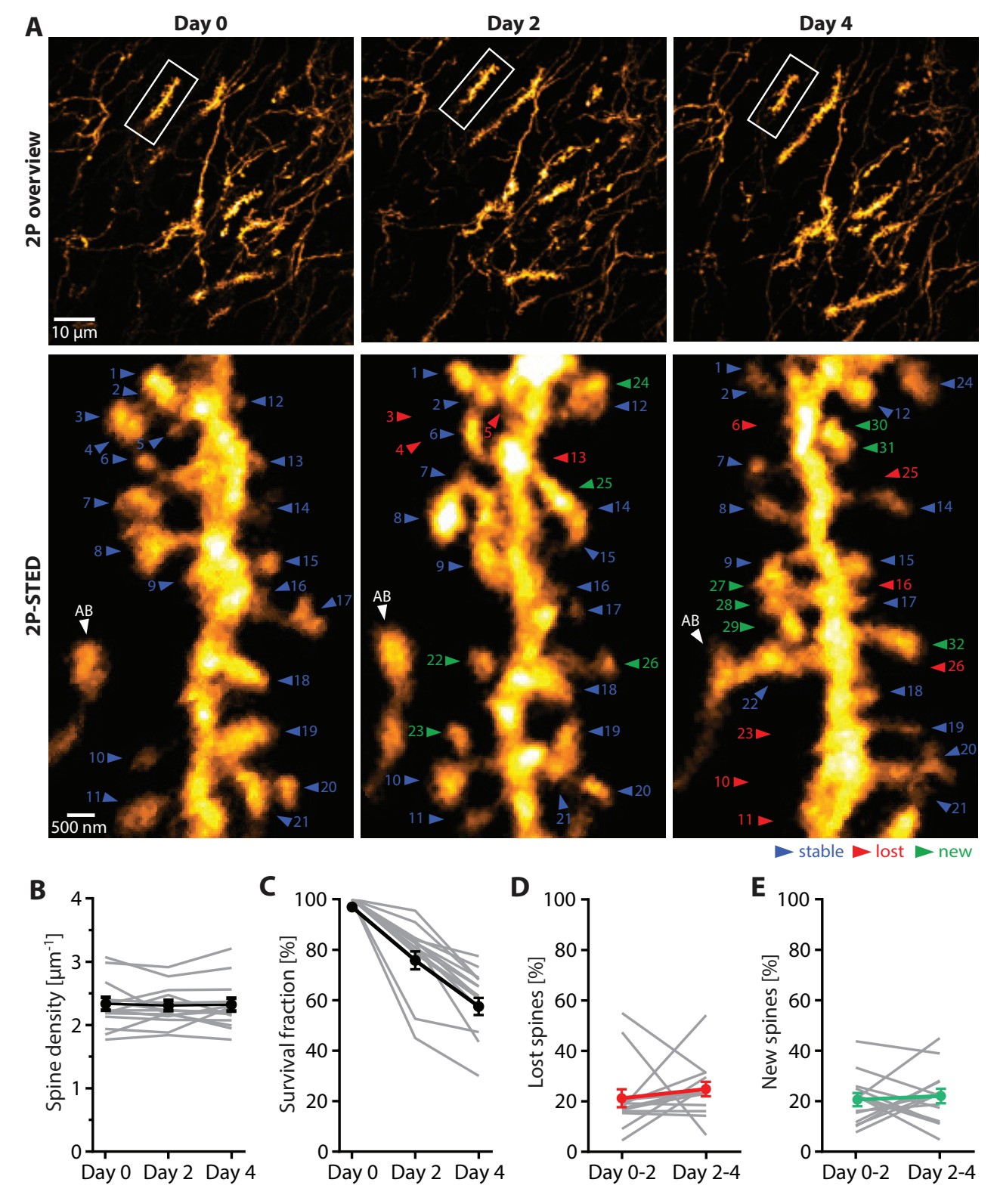

**Figure 3.** Turnover of spines on basal dendrites of CA1 pyramidal neurons in vivo. (**A**) Repetitive imaging of basal dendrites in CA1 area using 2P-STED microscopy. The upper panel shows low-magnification overviews containing the dendrite of interest highlighted with a white box. The lower panel shows the corresponding 2P-STED images of the dendrite over time. The images represent single z-planes. Dendritic spines with blue arrowheads were stable between imaging sessions. Red arrowheads mark lost spines, and green arrowheads mark new ones. The axonal bouton (AB) is marked by a

*Figure 3 continued on next page*

*Figure 3 continued*

white arrowhead. The numbering of spines is continuous. (B) Quantification of spine density over 4 days (n = 14 dendrites, 3 mice). (C) Quantification of the 4-day survival fraction of dendritic spines. (D) Fraction of lost spines and (E) fraction of new spines. Thin grey lines represent the measurements of single dendrites.

DOI: https://doi.org/10.7554/eLife.34700.014

The following source data and figure supplements are available for figure 3:

**Source data 1.** Source data of the parameters underlying *Figure 3* extracted from the turnover data set.

DOI: https://doi.org/10.7554/eLife.34700.017

**Source data 2.**

DOI: https://doi.org/10.7554/eLife.34700.018

**Source data 3.** Data for *Figure 3—figure supplement 1*.

DOI: https://doi.org/10.7554/eLife.34700.019

**Figure supplement 1.** 2P vs 2P-STED measurement of spine turnover in vivo.

DOI: https://doi.org/10.7554/eLife.34700.015

**Figure supplement 2.** 2P-STED time-lapse imaging of hippocampal dendrites in vivo.

DOI: https://doi.org/10.7554/eLife.34700.016

(*Figure 3E*). These results indicate that the rates of spine loss and gain were balanced and did not change over the sessions, which is consistent with the constant spine density we observed.

By comparison, spine turnover was less visible in 2P mode, amounting to a survival fraction of 74.9 ± 2.2% after 4 days (*Figure 3—figure supplement 1*).

## Spine turnover affects primarily small spines

Given the detailed morphological information in the 2P-STED images, we examined if there was a relationship between spine turnover and morphology. In cortex, spines with a large head have been shown to be more stable than filopodial and thin spines (*Holtmaat and Svoboda, 2009*; *Knott et al., 2006*), but it is unknown whether such a link also exists for the hippocampus.

After deconvolution of the images and 3D reconstruction (see Materials and methods), we analyzed the morphology of all spines that were visible on any of the imaging sessions (i.e. on days 0, 2 and/or 4; *Figure 4*), dividing the spines into two groups according to their 'observed persistence'. Spines that were visible on all three imaging sessions (Persistence >2 days) had larger heads than spines that were visible on only one or two sessions (Persistence ≤2 days), suggesting that small spines are more short-lived than large ones (>2 days: 0.03 $\mu m^3$ and 0.01–0.06 $\mu m^3$ versus ≤ 2 days: 0.02 $\mu m^3$ and 0.01–0.04 $\mu m^3$, median and interquartile range; *Figure 4A,B*; p<0.0001).

Refining this analysis, we plotted key morphological parameters (dendritic spine length, maximum head diameter, ratio of mean head to neck diameters) in three dimensions. The 3D plot revealed a clear difference in the distributions of the morphological parameters depending on the observed persistence of the spine (*Figure 4C*). A cluster analysis ('Agglomerative Hierarchical Clustering' analysis based on Euclidian distance in the parameter space, see Materials and methods) revealed at least three distinct populations of spines (*Figure 4—figure supplement 1*), which resemble the different categories commonly used in the literature to classify dendritic spines ('small' ≅ cluster 1, 'thin' ≅ cluster 2 and 'mushroom-like' ≅ cluster 3; *Figure 4D*). The latter category is generally composed of long spines with high head to neck diameter ratios. Notably, whereas around 50% of more persistent spines exhibited a mushroom-like morphology, less persistent spines rarely (7%) belonged to this category (*Figure 4E*). Even within cluster 3, less persistent spines had on average smaller heads and lower head to neck diameter ratios (*Figure 4F*).

## Discussion

We established chronic super-resolution imaging of dendritic spines in the intact hippocampus of living mice. It is based on a home-built upright 2P-STED microscope (*Bethge et al., 2013*; *Ter Veer et al., 2017*) equipped with a long working distance water immersion objective and a 'hippocampal window' to reach this deeply embedded brain structure (*Gu et al., 2014*; *Dombeck et al., 2010*).

STED microscopy has been used before for imaging dendritic spines in vivo, but only in superficial cortical layers and for one-off and acute imaging sessions (*Berning et al., 2012*; *Willig et al., 2014*).

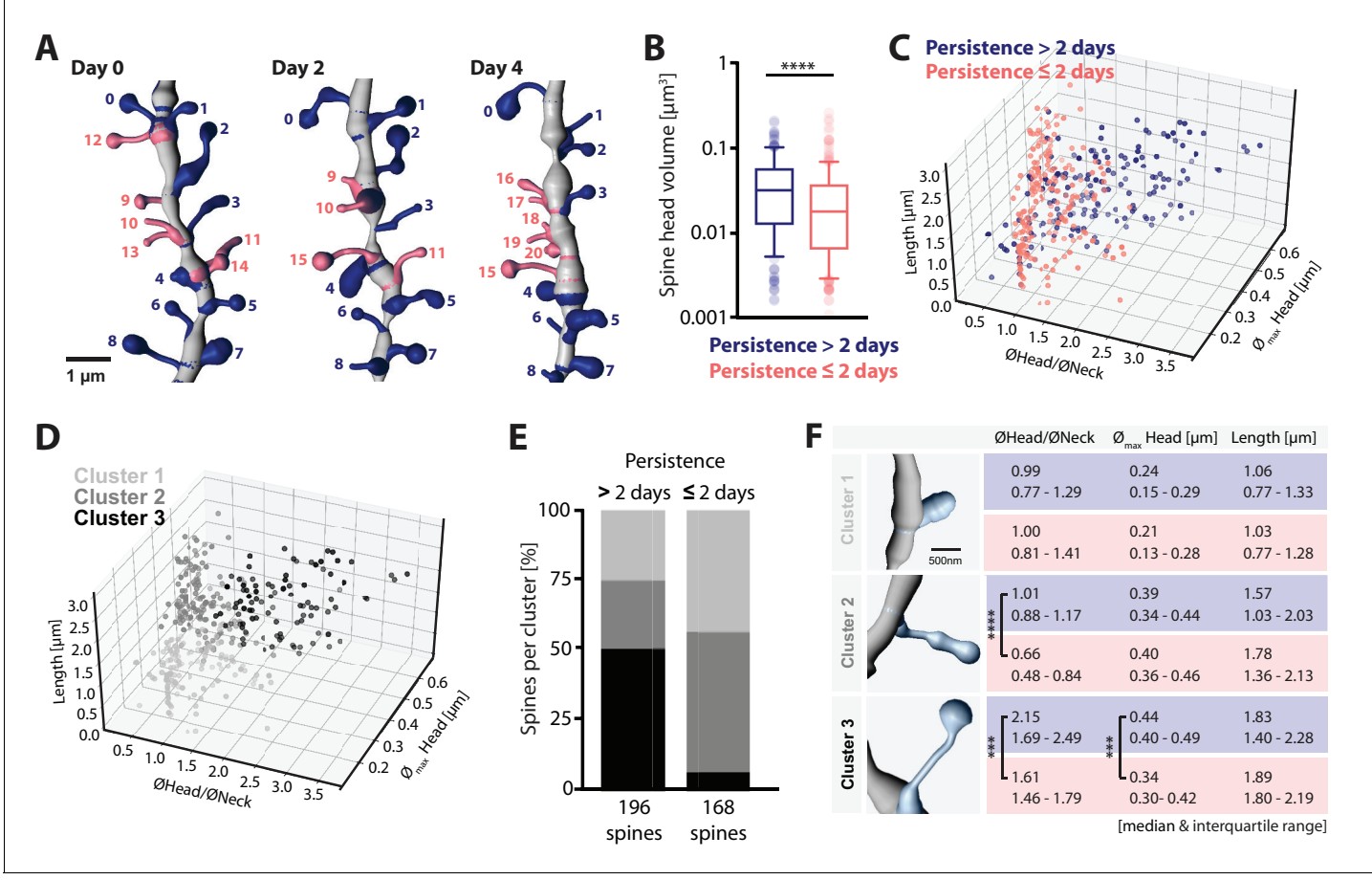

**Figure 4.** Structure-dynamics relationship of hippocampal spines. (A) 3D reconstruction of a dendrite imaged on days 0, 2 and 4. Spines persisting for more than 2 days (#0–8, blue), and 2 days or less (#9–20, salmon) are illustrated. (B) Spine head volumes measured on reconstructed dendrites (p<0.0001, Mann-Whitney test; n = 14 dendrites, 3 mice; box plot shows median and 10, 25, 75 and 90th percentiles). (C, D) 3D morphology plots visualizing the populations of spines observed persistent for more than 2 days and 2 days or less (C), and their affiliation to identified clusters 1, 2 and 3 (D) (see also *Figure 4—figure supplement 1A*). Plotted are, the ratio of mean head to neck diameters (ØHead/ØNeck), spine length and maximum head diameter (Ø$_{max}$ Head). (E) Quantification of spine proportions within identified clusters, distinguishing spines of different persistence (>2 days versus ≤2 days). (F) Table summarizing the morphological parameters utilized for cluster analysis: ØHead/ØNeck, Ø$_{max}$ Head and length of spines, for spines that persist for >2 days (blue) and ≤2 days (salmon). Data are represented as median and interquartile range (25th–75th percentile). Significant differences are marked by asterisks (***p<0.001, ****p<0.0001, unpaired t-test or Mann-Whitney test; for all comparisons see *Figure 4—figure supplement 1B*; n = 14 dendrites, 3 mice).

DOI: https://doi.org/10.7554/eLife.34700.020

The following source data and figure supplement are available for figure 4:

**Source data 1.** Underlying data for *Figure 4*.

DOI: https://doi.org/10.7554/eLife.34700.022

**Figure supplement 1.** Cluster analysis of spine morphology.

DOI: https://doi.org/10.7554/eLife.34700.021

Considering the importance of the hippocampus for learning and memory, it is of great interest to develop and improve approaches for faithful detection and monitoring of dendritic spines there.

Dendritic spines commonly serve as a morphological proxy for excitatory synapses, allowing inferences on synaptic strength and functional connectivity, for instance when their shape, size or number change over the course of an electrophysiological or behavioral experiment (*Yuste and Bonhoeffer, 2004*; *Holtmaat and Svoboda, 2009*). However, since spines have nanoscale dimensions and are densely packed in light scattering tissue, an accurate and quantitative anatomical readout is oftentimes difficult to obtain (*Bethge et al., 2013*).

Spine density has been shown to vary between 2 and 4 $\mu m^{-1}$ on proximal apical dendrites in *stratum radiatum* in adult rats (*Harris et al., 1992*; *Harris and Stevens, 1989*) and mice (*Bloss et al., 2018*) according to serial section transmission EM, and between 2 and 3 $\mu m^{-1}$ on basal oblique dendrites in *stratum oriens* in adult mice according to array tomography (*Bloss et al., 2016*). By comparison, the spine densities we detected in *stratum oriens* lie within the range of these measurements, while the two recent in vivo 2P microscopy studies reported values less than half the amount (*Gu et al., 2014*; *Attardo et al., 2015*).

As our STED approach only improved the lateral resolution, spines that protruded into the axial direction were still difficult to detect, explaining why our density values are still below those reported by EM. Indeed, correcting our data based on a simple geometric model removed the residual difference. In the near future, our method stands to benefit from ongoing technical advances, for instance in 3D beam shaping, which will improve axial resolution (*Gould et al., 2012*; *Patton et al., 2016*), and should enable nearly complete spine detection in vivo.

We could only image spines in the *stratum oriens* within 20 μm from the cover slip; beyond this distance, image quality degraded rapidly. This depth limitation was likely due to optical aberrations caused by mismatches in refractive index between the immersion medium, the glass cover slip and the sample (n ≈ 1.37 for brain tissue [*Lue et al., 2007*]). Unlike other objectives we have used for STED microscopy in brain tissue (*Bethge et al., 2013*; *Urban et al., 2011*), the new objective did not have a correction collar to reduce spherical aberrations, which otherwise probably would have permitted greater imaging depth. However, the use of adaptive optics is bound to allow for deeper imaging (*Ji, 2017*) and thus to reach dendrites in *stratum radiatum* and other areas of the hippocampus.

We suspect that the near total absence of motion artifacts in the images was primarily due to two effects: firstly, the implanted metal tube probably stabilized the brain mechanically, and secondly, blood pulsations are likely much weaker in the hippocampus where the vasculature is mostly formed by capillaries and has fewer large arteries than in superficial cortex (*Marinković et al., 1992*).

Faithful and complete detection of spines across space and time is absolutely critical for an accurate assessment of spine turnover. Failure to distinguish closely spaced spines will lead to an underestimate of spine turnover, because events of individual spines appearing or disappearing within a merged cluster will inevitably be missed, giving a false impression of spine stability (*Attardo et al., 2015*). Conversely, temporal fluctuations, for instance when spines rotate in or out of the optical axis, will lead to an overestimate of spine turnover. Moreover, biased detection of large spines might distort the measurements of spine turnover as large spines are reportedly less structurally plastic than small spines (*Matsuzaki et al., 2004*), as we have also shown here.

The spine detection problem was highlighted by two recent studies that pioneered chronic imaging in the hippocampus in vivo (*Gu et al., 2014*; *Attardo et al., 2015*) and reported very different lifetimes for spines on apical and basal dendrites of CA1 pyramidal neurons. The study by Gu *et al.* (*Gu et al., 2014*) reported that ~96% of spines survived for at least 16 days in *stratum radiatum*, whereas the study by Attardo *et al.* (*Attardo et al., 2015*) predicted that the entire population of spines in *stratum oriens* turned over within a month.

Both studies used 2P microscopy with relatively low NA optics and presented images of comparable quality, reporting similar average spine densities (~1.1 $\mu m^{-1}$) that were stable over time. While Gu et al. assessed spine turnover directly from the 2P images, Attardo et al. went a step further and used mathematical modeling to account for the effect of merged spines on apparent spine turnover. Importantly though, without applying the mathematical correction, their survival fraction was around 80% after 21 days, indicating that direct analysis of 2P images is highly problematic when it comes to establishing spine turnover rates.

Our super-resolution approach documented a spine survival fraction of 60% after 4 days, providing direct and unambiguous evidence that spines can turn over extremely rapidly in CA1 *stratum oriens*, supporting one of the main modeling-based conclusions of the Attardo *et al.* study. However, since we imaged just three time points over a relatively short period, it is hard to extrapolate our results in time and determine the extent to which the observed kinetics hold for the entire spine population. In fact, our observation that larger spines were more stable is a sign for the existence of multiple, kinetically distinct spine populations, unlike the Attardo *et al.* study, which argued for a single population.

We did not observe any overt signs of phototoxicity, such as dendritic 'blebbing' during short time-lapse sequences and across the imaging sessions (*Figure 3A*, *Figure 3—figure supplement 2*). This absence together with the fact that spine density remained constant indicates that the observed kinetics were real and not an artifact induced by our hippocampal window approach or 2P-STED imaging. The use of the NMDA-receptor antagonist ketamine could be problematic given the important role of NMDA-receptors in hippocampal synaptic plasticity. However, as the drug was applied only for a couple of hours per imaging session and we did not induce any plasticity, it is unlikely that our results were compromised by it. The use of more innocuous drugs or performing the experiments with non-anesthetized and head-fixed animals will be preferable in the future.

As we did not image in the *stratum radiatum*, we do not know to what extent the divergence in spine turnover between *stratum radiatum* reported by Gu *et al.* and *stratum oriens* (Attardo *et al.* and our data) reflects methodological or genuine anatomical or physiological differences. In fact, *stratum radiatum* primarily receives input from CA3 (*Somogyi, 2010*; *Cappaert et al., 2014*), whereas *stratum oriens* also has afferents from entorhinal cortex and amygdala. In addition, there may be intrinsic differences in the postsynaptic neuron, which account for dendrite-specific spine turnover.

Our finding that hippocampal dendrites are subject to ongoing intense anatomical remodeling supports the view of the hippocampus as a highly dynamic structure designed to encode and process new memories, but not as a long-term repository of information (*Frankland and Bontempi, 2005*). The situation may be quite different in cortical areas, where typically a large fraction of spines is stable for many weeks in adult mice (*Grutzendler et al., 2002*; *Majewska et al., 2006*; *Trachtenberg et al., 2002*), unless they are subjected to behavioral training (*Xu et al., 2009*; *Yang et al., 2009*) or sensory manipulations (*Hofer et al., 2009*; *Keck et al., 2008*). However, whether and how spine turnover actually reflects memory-relevant functional adaptations in synaptic strength and connectivity remains to be determined.

In summary, the present work adds up to a substantial advance for the study of hippocampal synapses in living mice, extending the scope of super-resolution microscopy to a deeply embedded brain structure that is critical for memory function. By correlating spine-level structural changes with genetic, molecular and behavioral interventions and assays, our chronic super-resolution imaging approach creates manifold opportunities to study the neurobiological mechanisms and functional significance of spine plasticity in the mammalian brain.

## Materials and methods

### Animals

We used adult female and male transgenic mice (*Thy1*-H$^{YFP/+}$ and *Thy1*-M$^{GFP/+}$, 4–12 months old) where a subset of pyramidal neurons is fluorescently labeled (*Feng et al., 2000*). The mice were group housed by gender at a day/night cycle of 12/12 hr. All procedures were in accordance with the Directive 2010/63/EU of the European Parliament and approved by the Ethics Committee of Bordeaux and by the government of North Rhine Westphalia.

### Hippocampal window surgery

Chronic hippocampal windows were implanted as described previously (*Gu et al., 2014*; *Schmid et al., 2016*), providing optical access to the *stratum oriens* of the CA1 region of the hippocampus (*Figure 1B*). In brief, mice were anesthetized with an i.p. injection of ketamin/xylazine (0.13/0.01 mg/g bodyweight) and received subcutaneous injections of analgesics (buprenorphine, 0.05 mg/kg) and anti-inflammatory agents (dexamethasone, 0.2 mg/kg) to prevent the brain from swelling during the surgical procedure. Using a dental drill, a craniotomy of 3 mm in diameter was performed above the right hemisphere (stereotactic coordinates: anteroposterior, −2.2 mm; mediolateral, +1.8 mm relative to bregma). The dura and somatosensory cortex above the hippocampus were carefully aspirated, while leaving the external capsule of the hippocampus intact. Subsequently, a custom-made metal tube sealed with a coverslip on the bottom side (both 3 mm in diameter, height 1.5 mm and 0.13 mm) was inserted into the craniotomy and fixed to the skull with dental acrylic. Following the surgery, mice received analgesics for 3 days (buprenorphine, 0.05 mg/kg, s.c.) and were allowed to recover from the surgery for 4 weeks before experiments started.

## Two-photon STED microscopy

We used a custom-made upright 2P-STED microscope (*Figure 1A*) based on two-photon excitation and stimulated emission depletion (STED) using pulsed lasers (*Bethge et al., 2013*; *Ter Veer et al., 2017*). Briefly, a femtosecond mode-locked Ti:Sapphire laser (Chameleon, Coherent, Santa Clara, CA) operating at 80 MHz and emitting light at 834 nm was used in combination with an optical parametric oscillator (OPO BASIC Ring fs, APE, Berlin, Germany) to produce STED light pulses at 598 nm with ~150 fs pulse duration. The pulses were stretched to ~100 ps by passing them through a glass rod and a 20 m long polarization-maintaining fiber. A STED light reflection served to synchronize a second Ti:Sapphire laser (Tsunami, Spectra Physics, Darmstadt, Germany) tuned to 910 nm and running at a repetition rate of 80 MHz, which was used for two-photon excitation of the fluorophores. Synchronization and optimal pulse delay were achieved with phase-locked loop electronics (3930, Lok-to-Clock, Spectra Physics). The STED doughnut was created by passing the STED beam through a vortex phase mask (RPC Photonics, Rochester, NY), which imposed a helical $2\pi$-phase delay on the wave front. Wave plates ($\lambda/2$ and $\lambda/4$) were used to make the STED light circularly polarized before it entered into the objective. The 2P and STED beam were combined using a long-pass dichroic mirror. The two laser beams were moved over the sample in all three dimensions using a galvanometric x-y scanner (Janus IV, TILL Photonics) combined with a z-focusing piezo actuator (Pifoc, PI, Karlsruhe, Germany). To bridge the physical distance between the surface of the brain and the deeply embedded hippocampus, we employed a long-working distance water-immersion objective (Nikon CFI Apo 60X W NIR, 1.0 NA, 2.8 mm WD). The epi-fluorescence was de-scanned and imaged onto an avalanche photodiode (SPCM-AQR-13-FC, PerkinElmer, Villebone-sur-Yvette, France). Signal detection and peripheral hardware were controlled by the Imspector scanning software (Abberior Instruments, Göttingen, Germany) via a data acquisition card (PCIe-6259, National Instruments). Optical resolution was assessed by imaging 40 nm fluorescent nanospheres (yellow-green fluospheres, Invitrogen) immobilized on glass slides (*Figure 1C*). Regions of interest were consecutively imaged in 2P and 2P-STED mode (10 × 10 $\mu m^2$; 10 nm pixel size; 50 $\mu s$ pixel dwell time). The laser power at the sample was typically around 5–20 mW for 2P and 5–15 mW for STED.

## In vivo imaging

Mice were anesthetized with an i.p. injection of ketamin/xylazine (0.13/0.01 mg/g bodyweight) and the eyes were protected with ointment (Bepanthen). The mouse was fixed to a custom-made stereo-tactic frame and kept at body temperature using a heating pad. For the quantification of spine density, 2P-STED and 2P images of basal dendrites of CA1 pyramidal neurons were acquired applying identical acquisition parameters (10 × 10 × 4–8 $\mu m^3$; 20–40 nm pixel size; 0.5–1 $\mu m$ z-step; 20–70 $\mu s$ pixel dwell time). For the chronic repetitive imaging, the position of the field of view (FOV) was registered in the first imaging session with the help of vascular landmarks and cell bodies of CA1 pyramidal neurons. This allowed for subsequent retrieval of the FOV for each mouse. An overview 2P image z-stack (100 × 100 × 20 $\mu m^3$; 200 nm pixel size; 1 $\mu m$ z-steps) was acquired at each time point starting from the coverslip. Subsequently, identified distal stretches of basal dendrites in CA1 *stratum oriens* (5–20 $\mu m$ in depth) were imaged in 2P and 2P-STED mode. A single imaging session lasted for ~1 hr and mice woke up in their home cage afterwards. All mice survived the imaging sessions and recovered normally from the anesthesia. Mice were excluded from the analysis if the hippocampal window was faulty or if their fluorescence levels were too low.

## Image analysis

The 2P and 2P-STED images were spatially filtered (1 pixel median filter) and their brightness and contrast were individually adjusted. Optical resolution of the 2P-STED microscope was assessed by taking images of 40 nm fluorescent nanospheres. We measured two-pixel line profiles across the nanospheres and perpendicular to the spine neck, and fitted them with a Lorentzian function, whose full-width at half maximum (FWHM) served as a measure of the spatial resolution or the neck width, respectively. Paired spine neck measurements were only done for spines where the necks could be discerned in both the 2P and 2P-STED images. Spine density was determined as described before (*Gu et al., 2014*; *Fuhrmann et al., 2007*; *Holtmaat et al., 2005*). All dendritic spines, which extended out laterally from the dendritic shaft by more than 200 nm, were counted by manually

scrolling through the z-stacks. Spine density was calculated as the number of spines divided by the dendritic length in micrometer (*Figure 2—figure supplement 1*). For the 4-day repetitive imaging data set, the spine density of each dendrite was determined blindly with respect to the time point of acquisition. Lost and new spines over the 4-day interval were identified by scrolling through the z-stacks in a chronological order. Spines were scored lost, if they were under the 200 nm threshold. Spines were considered new, if their position on the dendrite relative to neighboring spines shifted by $\geq$500 nm (*Gu et al., 2014*), (*Holtmaat et al., 2005*). The survival fraction of spines ($F_s$) was calculated as the number of remaining spines at day t ($N_r(t)$) divided by the number of spines at day 1 ($N(1)$), expressed in percent:

$$F_s = \frac{N_r(t)}{N(1)} \times 100 \tag{1}$$

The fraction of lost spines ($F_{lost}$) was assessed for each time point by dividing the number of spines that disappeared on day t ($N_L(t)$) by the total number of spines ($N(t)$):

$$F_{lost} = \frac{N_L(t)}{N(t)} \times 100 \tag{2}$$

The fraction of new spines ($F_{new}$) was assessed for each time point by dividing the number of spines that appeared on day t ($N_n(t)$) by the total number of spines ($N(t)$):

$$F_{new} = \frac{N_n(t)}{N(t)} \times 100 \tag{3}$$

## Reconstruction of dendritic segments

To enhance contrast, raw 2P-STED image stacks were subjected to deconvolution (Huygens HuCore version 17.4, Scientific Volume Imaging b.v., Hilversum, The Netherlands) utilizing a theoretical PSF based on microscope parameters and classic maximum likelihood estimation (cmle) with a quality stop criterion of 0.01, automatic background estimation and a signal-to-noise ratio (SNR) of 15.

After deconvolution, image stacks were 3D reconstructed in IMARIS v 8.4.1 (Bitplane AG, Zurich). Dendrites and associated spines were reconstructed semi-automatically using the Filament Tracer module. Morphological parameters of spines (mean head width, mean neck width, maximum head width, spine head volume and spine length) were measured and exported for further analysis. To quantify persistence, spines were manually traced over time by assigning individual spines of consecutive imaging sessions to each other. Morphological parameters of spines that were present in all three imaging sessions (persistence >2 days) were measured on the last imaging time point. Morphological parameters of spines that were observed for 2 days or less (persistence $\leq$2 days) were measured at the time point of first appearance (new spines), or at the last time point of presence (lost spines).

## Cluster analysis

For the cluster analysis, the spine parameters, ratio of the mean head to neck diameters (ØHead/ØNeck), the maximum head width (Ø$_{max}$ Head) and spine length were taken into account. Agglomerative Hierarchical Clustering Analysis (AHC) was performed using Python's scikit-learn toolbox. Before cluster analysis, the parameters were centered and scaled to unit variance using the standard scaler from Python's scikit-learn toolbox. AHC dissimilarity level was calculated based on Euclidean distance. Agglomeration was performed using Ward's method. By fitting the hierarchical clustering to the population of all analyzed spines, we identified three clusters.

## Tissue preparation and immunochemistry

All mice were injected with a lethal dose of pentobarbital (200 mg/kg, i.p., Centravet) and perfused transcardially with saline solution followed by 4% (w/v) paraformaldehyde in 0.1 M Phosphate buffer. Brains were removed, post-fixed in 4% paraformaldehyde for 6–8 hr, and then sectioned at 40 µm in the coronal plane on a vibratome (VT1200, Leica). Free floating sections were blocked and permeabilized with a blocking buffer containing 5% normal goat serum (NGS) and 0.5% TritonX-100 diluted in PBS for 1 hr at room temperature. They were then incubated with the rabbit anti-GFP (polyclonal serum 1/1000, Invitrogen) diluted in PBS containing 0.1% tween and 1% NGS for 24 hr at 4°C; and

with the photostable Atto647N-conjugated goat anti-rabbit secondary antibody for 45 min at room temperature (2 µg/ml, Sigma-Aldrich). The sections were mounted directly on coverslips (high-performance coverglass D = 0.17 ±0.005 mm refractive index = 1.526, Zeiss) using Mowiol (Mowiol 4–88 Calbiochem #475904, refractive index = 1.460) for imaging.

## Fixed tissue imaging

STED images of fixed brain slices were acquired on a commercial STED microscope (Leica DMI6000 TCS SP8 X, Leica Microsystems, Mannheim, Germany), using a 93X glycerol objective with a numerical aperture of 1.3 that was equipped with a motorized correction collar. The microscope was supplied with a white light laser 2 (WLL2) with freely tunable excitation from 470 to 670 nm. The STED module used a pulsed laser for depletion at 775 nm. Image stacks of basal dendrites of CA1 pyramidal neurons were acquired with a pixel size of 20 nm, a z-step size of 200 nm and at a scan speed of 200 Hz using three line averages and four frame accumulations. Spine density analysis was performed as described for the in vivo image analysis (see Image Analysis).

## Quantification and statistics

Quantifications and statistical analysis were performed using Microsoft Excel and GraphPad Prism 7 (GraphPad Software, Inc.). All data were tested for normal distribution using the Shapiro-Wilk normality test. For normally distributed data t-tests, either paired (data of *Figures 1–3*) or unpaired (*Figure 4F*), were performed to test for statistical significance. Not normally distributed data were tested with Mann-Whitney test (*Figure 4B*; *Figure 4F*; *Figure 4—figure supplement 1B*). All values are represented as mean ± sem, unless stated otherwise.

## Acknowledgements

This work was supported by the DZNE, grants from the Deutsche Forschungsgemeinschaft (SFB1089; C01 and B06), CoEN (CoEN3018) and the EU (ERA-NET MicroSynDep) to MF, the Fondation pour la Recherche Médicale (DEQ20160334901) and CoEN (ANR-16-COEN-0003–02) to UVN. TP was supported by a Boehringer Ingelheim Fonds PhD fellowship and a PhD extension grant from the Fondation pour la Recherche Médicale (634FDT20160435677). We thank Mirelle ter Veer for work on the STED microscope, Steffen Burgold for his support with IMARIS, all lab members for comments on the manuscript and the DZNE Light Microscope Facility (LMF), Image and Data Analysis Facility (IDAF) and Animal Research Facility (ARF) for technical support. Fixed tissues imaging was performed at the Bordeaux Imaging Center (BIC), a service unit of the CNRS-INSERM and Bordeaux University, member of the national infrastructure France BioImaging. We thank Patrice Mascalchi (BIC) for technical support.

## Additional information

### Funding

| Funder | Grant reference number | Author |
| --- | --- | --- |
| Fondation pour la Recherche Médicale | DEQ20160334901 | U Valentin Nägerl |
| COEN | ANR-16-COEN-0003-02 | U Valentin Nägerl |
| COEN | CoEN3018 | Martin Fuhrmann |
| Deutsche Forschungsgemeinschaft | SFB1089 | Martin Fuhrmann |
| ERA-Net | MicroSynDep | Martin Fuhrmann |
| Boehringer Ingelheim Fonds | Graduate Student Fellowship | Thomas Pfeiffer |
| Fondation pour la Recherche Médicale | Graduate Student Fellowship 634FDT20160435677 | Thomas Pfeiffer |

The funders had no role in study design, data collection and interpretation, or the decision to submit the work for publication.

## Author contributions

Thomas Pfeiffer, Formal analysis, Investigation, Methodology, Writing—original draft, Writing—review and editing; Stefanie Poll, Formal analysis, Investigation, Writing—review and editing; Stephane Bancelin, Julie Angibaud, VVG Krishna Inavalli, Investigation, Methodology, Writing—review and editing; Kevin Keppler, Resources; Manuel Mittag, Software, Methodology, Writing—review and editing; Martin Fuhrmann, Conceptualization, Supervision, Writing—original draft, Project administration, Writing—review and editing; U Valentin Nägerl, Conceptualization, Supervision, Funding acquisition, Writing—original draft, Project administration, Writing—review and editing

## Author ORCIDs

Thomas Pfeiffer 
Stefanie Poll 
Stephane Bancelin 
Martin Fuhrmann 
U Valentin Nägerl 

## Ethics

Animal experimentation: All procedures of this study were in accordance with the Directive 2010/63/EU of the European Parliament and approved by the Ethics Committee of Bordeaux and by the government of North Rhine Westphalia.

## Decision letter and Author response

Decision letter https://doi.org/10.7554/eLife.34700.028
Author response https://doi.org/10.7554/eLife.34700.029

# Additional files

## Supplementary files

• Transparent reporting form
DOI: https://doi.org/10.7554/eLife.34700.023

## Data availability

Source data files have been provided for Figures 1, 2, 3 & 4 and Figure 1-figure supplement 1 and Figure 3-figure supplement 1.

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
