## [Decision Letter]

Thank you for submitting your article "Chronic STED imaging reveals high turnover of dendritic spines in the hippocampus in vivo" for consideration by *eLife*. Your article has been reviewed by 3 peer reviewers, including Karel Svoboda as the Reviewing Editor, and the evaluation has been overseen by Eve Marder as the Senior Editor. The following individual involved in review of your submission has also agreed to reveal their identity: Anthony Holtmaat (Reviewer #2).

The reviewers have discussed the reviews with one another and the Reviewing Editor has drafted this decision to help you prepare a revised submission.

Imaging and tracking of dendritic spines in vivo poses two distinct challenges: 1. the resolution, contrast and sensitivity need to be sufficiently high to discern spines from the parent dendrite and from one another. 2. dealing with motion artifacts, which can equally degrade the resolution. The first challenge is particularly important in hippocampus, since this is a deep structure with high spine densities. The latter is usually more of an issue in neocortex because it contains a high density of large, pulsating, blood vessels. 2-photon laser scanning microscopy allows imaging dendritic microstructures over time, however its resolution is limited and may not always be sufficient to resolve/detect all spines in the hippocampus. Modeling approaches have been applied to infer dynamics and turnover based on 2P in vivo images (Attardo et al., 2015).

The authors have applied 2P-STED, a super-resolution method, to image spine dynamics in the mouse hippocampus in vivo over 4 days. They dealt with access and motion artifacts by surgically replacing the cortex with a metal cylinder. To showcase the possibilities of the microscope and prep the authors provide paired comparisons between regular 2-photon and 2-photon STED images and measurements of spines. Then they went on to show that, due to the superior resolution, spine densities are closer to the values reported in EM (used as ground truth). The spine turnover rates are higher than what was reported in a previous 2-photon imaging study with a similar preparation (Gu et al. 2014), and similar to the modeling-based measurements derived from microendoscopic images (Attardo et al. 2015).

This paper describes a proof of principle using 2P-STED in vivo and provides a first analysis of structural plasticity data. We have no major concerns as to the validity of the results or their interpretation, but suggest expanding the analysis in order to provide more details about the structure-dynamics relationships.

Major concerns:

1) The spine density reported with STED is still lower than that measured with EM. The authors should produce best-of-class STED images from fixed samples of the same mice and compare the spine densities, this would also control for possible effects of the surgery, loss of resolution in vivo etc.

2) The paper is short, but still a bit flabby. In parts it reads a bit like an infomercial for STED. The resolution advantage over 2p is obvious and should be mentioned once. The effects on measured spine density are also obvious and don't need to be belabored multiple times etc.

3) The data are under-analyzed. One of the main advantages of the current study as compared to the Attardo et al. study, is the information about spine structure (size, length, head size, neck size etc). In cortex, there are clear structure-dynamics relationships, but for hippocampus in vivo this is not known. The authors should expand their analysis into this direction and show whether such relationships between spine dynamics and spine length/size, head size etc exist.

4) The authors argue that the current data set experimentally confirms the modeling data from the Attardo et al. study. Indeed, the first 4 days of imaging suggest that the entire population of spines may be impermanent. However, due to the limited sampling rate, it is not inconceivable that the authors could not detect a more or less stable spine population. To get more insight into this, it would be interesting to analyze the survival rate of new spines, and perhaps model this to estimate the stability and turnover of the entire population. In cortex new spines tend have a half-life in the order of 2 days or so, but in the current data set they could probably live longer. This, together with the finding that about 30% of the entire population disappears over 4 days, may indeed suggest that the whole population of spines is subject to turnover (in contrast to cortex, where the turnover rates to a large extent concerns newly formed spines). The structure-dynamics relationship could also be incorporated into this model.

5) The authors need to discuss or analyze more extensively what were the critical factors in causing the quite different turnover rates between the current study and the Gu et al. paper (one of the co-authors is on both papers!). It is difficult to fathom that the different values (96% stability over 16 days in Gu et al., and 60% stability over 4 days in the current paper) were entirely due to the improved resolution or the hippocampal region. A direct comparison of turnover rates scored in 2P and 2P-STED could have possibly hinted at the underlying factors, or could have identified the type of spines that can be seen turning over in 2P-STED and not in 2P. Furthermore, they should also discuss if long-term 2P-STED could provide better data for cortex or not, given that cortex is usually more prone to movement artefacts?

6) Since this paper focuses heavily on showing the potential of in vivo 2P-STED, we suggest that the authors provide a more substantive comparison between 2P-STED and 2P images in 3D, that is by showing the image stacks that were typically used for scoring, in addition to the projections. This could be done as a figure supplement or as a new figure. This is important since spines are usually scored in 3D (as the authors do too), and resolving them in the axial dimension is most problematic – and probably still difficult in 2P-STED (as the authors comment). This might relate to the remaining difference between the reported spine densities in 2P-STED and EM.

7a) Most of the spine necks measured in 2P-STED are smaller than the PSF in 2P (as measured on beads). For example, there are spines that measure 500-700 nm in 2P, but less than 200 in 2P-STED. Wouldn't one expect the size of the smaller necks in 2P to be limited to the PSF of the microscope? Can the authors explain this phenomenon?

b) Can the authors also explain why the PSF in 2P is highly variable (the FWHM varies from ± 300-450nm)? We assume that the beads were imaged under optimal and standardized conditions. So, what causes the large variability? The PSF in the axial dimension should be presented as well, as this is the dominant factor in the difficulty to discern spines from the parent dendrite in 2P and probably also in 2P-STED.

c) Along similar lines since the PSF, it would be better to measure the PSFs on beads that are embedded in the preparation. This could change the variance and the bounds as seen in Figure 1C, as well as the relationship of the PSF to the measurements in E.

8) The authors should specify the length of dendrite that was analyzed, and provide the number of spines that were counted. For example, subsection “Dendritic spine density of CA1 pyramidal neurons in vivo”, first paragraph and subsection “Dendritic spines undergo high morphological turnover in vivo”, second paragraph.

9) The authors used a parametric test to report differences throughout. Did the authors check for normality of the data? They need to specify this.

10) All dendritic images presented in the manuscript appear saturated, which makes it difficult to judge the accuracy of spine counting and the health of dendrite.

11) The criteria used in counting spines are a bit vague. For example, in Figure 3A, day 1, spine 10 does not seem to be connected to the dendritic shaft; spine 9 seems to be a small bump from the dendritic shaft, whereas similarly-sized bumps (e.g., between spine 18 and 19) are not counted. The methods of counting need to be documented in great and quantitative detail.

---

## [Author Response]

1) The spine density reported with STED is still lower than that measured with EM. The authors should produce best-of-class STED images from fixed samples of the same mice and compare the spine densities. This would also control for possible effects of the surgery, loss of resolution in vivo etc.

Taking up your advice, we acquired STED images in fixed hippocampal brain slices obtained from the same transgenic mice we had used for the in vivo experiments. The spine density in these fixed samples approaches the values reported by EM. To understand the remaining difference, we used a geometrical model based on the PSF of our microscope and the dimensions of the measured dendrites. The model makes the point that the limited z-resolution of the STED approach is the main reason, why we still underreport spine density relative to EM. (Revised Figure 2; Figure 2—video 1.)

2) The paper is short, but still a bit flabby. In parts it reads a bit like an infomercial for STED. The resolution advantage over 2p is obvious and should be mentioned once. The effects on measured spine density are also obvious and don't need to be belabored multiple times etc.

We heeded the reviewer’s advice and eliminated the redundancies about the STED benefit.

*3) The data are under-analyzed. One of the main advantages of the current study as compared to the Attardo et al. study, is the information about spine structure (size, length, head size, neck size etc). In cortex, there are clear structure-dynamics relationships, but for hippocampus* in vivo *this is not known. The authors should expand their analysis into this direction and show whether such relationships between spine dynamics and spine length/size, head size etc exist.*

We thank the reviewers for raising this important point, which prompted us to extract more morphological information from our time-lapse data set to analyse spine structure-dynamics relationships.

We carried out two types of analyses, which indicate that a structure-dynamics relationship indeed exists for hippocampal spines in vivo, resembling the case in the cortex, where smaller spines generally appear to be less stable.

Firstly, we show that spines that were visible on only one or two days have on average smaller heads than spines that were visible on all three imaging sessions.

Secondly, we performed a mathematical cluster analysis of the morphological parameters, which revealed three distinct spine populations (which incidentally resemble the classical spine types: cluster 1 ≅ small; cluster 2 ≅ thin; cluster 3 ≅ mushroom-like). Whereas 50% of ‘persistent’ spines (i.e. that were visible on all three imaging sessions) appeared in cluster 3, spines that were visible only on one or two sessions rarely (7%) exhibited a mushroom-like morphology. (New Figure 4; new Figure 4—figure supplement 1.)

4) The authors argue that the current data set experimentally confirms the modeling data from the Attardo et al. study. Indeed, the first 4 days of imaging suggest that the entire population of spines may be impermanent. However, due to the limited sampling rate, it is not inconceivable that the authors could not detect a more or less stable spine population. To get more insight into this, it would be interesting to analyze the survival rate of new spines, and perhaps model this to estimate the stability and turnover of the entire population. In cortex new spines tend have a half-life in the order of 2 days or so, but in the current data set they could probably live longer. This, together with the finding that about 30% of the entire population disappears over 4 days, may indeed suggest that the whole population of spines is subject to turnover (in contrast to cortex, where the turnover rates to a large extent concerns newly formed spines). The structure-dynamics relationship could also be incorporated into this model.

As suggested, we examined the kinetic data to look for evidence of a more long-lived spine population. We analysed the behaviour of new spines (i.e. spines that appeared on day 2) and observed that 66% of them were still visible during the next imaging session (day 4), indicating that newly formed spines may have a longer half-life in hippocampus than in cortex.

To take a closer look at the possibility of a stable spine population, we extrapolated the fraction of surviving spines by using a simple kinetic model based on a geometrical progression. The number of surviving spines on day n+1 can be expressed as:

u_n+1_=u_n-_ F∗u_n_

where u_n_ is the number of surviving spines on day n and F is the fraction of lost spines from day n to n+1 among the surviving ones. Note that this definition of F differs from the lost fraction of spines, since it excludes the new spines on day n that are lost on day n+1. Therefore, the number of surviving spines is

u_n_=u_0_(1-F)^n^

Fitting the model to the in vivo data set, we estimated F = 22%, which corresponds to a spine half-life of 5.5 days. Moreover, the fit indicates that the survival fraction indeed converges to zero, which is inconsistent with the existence of a stable spine population. However, the ability of this modelling approach to identify kinetically distinct populations is at present not very good, given the limited sampling duration.

By comparison, fitting the model to the 2P data set, yields a spine half-life of 9 days (see Author response image 1). This confirms that data with lower spatial resolution indeed lead to an over-estimation of spine stability, as discussed in the Attardo study.

Because of the limited sampling and the inherent unreliability of longer-term extrapolations, we decided to omit this data from the manuscript. However, we discuss that these important kinetic issues need to be tested experimentally.

5) The authors need to discuss or analyze more extensively what were the critical factors in causing the quite different turnover rates between the current study and the Gu et al. paper (one of the co-authors is on both papers!). It is difficult to fathom that the different values (96% stability over 16 days in Gu et al., and 60% stability over 4 days in the current paper) were entirely due to the improved resolution or the hippocampal region. A direct comparison of turnover rates scored in 2P and 2P-STED could have possibly hinted at the underlying factors, or could have identified the type of spines that can be seen turning over in 2P-STED and not in 2P. Furthermore, they should also discuss if long-term 2P-STED could provide better data for cortex or not, given that cortex is usually more prone to movement artefacts?

In the revised manuscript we elaborate more about why the Gu et al. study reported a much higher spine stability. The reason is likely a combination of microscope resolution and hippocampal region. However, without a direct experimental comparison (our STED approach limited us to *stratum oriens),* we cannot provide a quantification for how much Gu et al. over-estimated stability versus how much spines are really more stable in the *stratum radiatum*.

Having said that, we did observe a considerable level of spine turnover even with just 2P, which one might think supports the case of a biological difference between basal and apical CA1 dendrites. However, our 2P approach, which benefited from a higher objective NA, already detected about 45% more spines than Gu et al., so it is not a fair comparison, and thus the same argument about over-estimating spine stability applies. We have tried to address these questions openly in the Discussion.

Regarding the point about the utility of 2P-STED for cortex, it is likely that improved spatial resolution will still provide better data, even if imaging in superficial brain areas is more prone to movement artefacts. The benefit of STED will be less obvious in as much as sample motion and not optics is the dominant factor behind image blur. (New Figure 3—figure supplement 1.)

6) Since this paper focuses heavily on showing the potential of in vivo 2P-STED, we suggest that the authors provide a more substantive comparison between 2P-STED and 2P images in 3D, that is by showing the image stacks that were typically used for scoring, in addition to the projections. This could be done as a figure supplement or as a new figure. This is important since spines are usually scored in 3D (as the authors do too), and resolving them in the axial dimension is most problematic – and probably still difficult in 2P-STED (as the authors comment). This might relate to the remaining difference between the reported spine densities in 2P-STED and EM.

Indeed, resolving dendritic spines along the optical axis is still difficult, as our new approach did not provide an improved z-resolution. To comply with the reviewer’s suggestion, we provide a new supplementary figure that illustrates how we scored spines. In addition, as mentioned above, we have used a geometrical model to estimate the effect of limited z-resolution on spine counting. Taking this fraction of invisible spines into account, the estimated spine density closely matches the reported spine densities reported by Bloss et al. 2018 and Harris et al. 1992 using EM. (New Figure 1—figure supplement 1; new Figure 2—figure supplement 1.)

7a) Most of the spine necks measured in 2P-STED are smaller than the PSF in 2P (as measured on beads). For example, there are spines that measure 500-700 nm in 2P, but less than 200 in 2P-STED. Wouldn't one expect the size of the smaller necks in 2P to be limited to the PSF of the microscope? Can the authors explain this phenomenon?

We thank the reviewers for pointing out this inconsistency. Revisiting the 2P images, we realized that some of our measurements were contaminated by instances where spines appeared merged in the images with neighbouring spines or other poorly resolved structures due to the lower spatial resolution, yielding artefactually wider neck measurements. By carefully looking at the corresponding 2P-STED images, we could eliminate these cases. In addition, we have now excluded measurements where the R^2^ of the fit for the line profiles was below 0.8. (Revised Figure 1E.)

b) Can the authors also explain why the PSF in 2P is highly variable (the FWHM varies from ± 300-450nm)? We assume that the beads were imaged under optimal and standardized conditions. So, what causes the large variability?

We also thank the reviewers for spotting this error, due to the inclusion of clustered beads in the analysis. Removing these (based on checking the corresponding 2P-STED image) and applying the threshold described above (fits with R^2^ < 0.8 were excluded), the variability was greatly reduced. We obtained a value of 325 nm ± 5 nm for 2P, which is very close to the theoretical value of 320 nm for a 1.0 NA objective and 900 nm excitation. (Revised Figure 1C.)

The PSF in the axial dimension should be presented as well, as this is the dominant factor in the difficulty to discern spines from the parent dendrite in 2P and probably also in 2P-STED.

We now present the PSF in the axial dimension as requested. (New Figure 1—figure supplement 1.)

c) Along similar lines since the PSF, it would be better to measure the PSFs on beads that are embedded in the preparation. This could change the variance and the bounds as seen in Figure 1C, as well as the relationship of the PSF to the measurements in E.

We agree with the reviewers that measuring PSFs in situ would be the preferred strategy, but unfortunately for practical reasons we cannot fulfil this request at this stage.

8) The authors should specify the length of dendrite that was analyzed, and provide the number of spines that were counted. For example, subsection “Dendritic spine density of CA1 pyramidal neurons in vivo”, first paragraph and subsection “Dendritic spines undergo high morphological turnover in vivo”, second paragraph.

We now provide a table containing the source data of the spine turnover analysis. (New Figure 3—source data 1.)

9) The authors used a parametric test to report differences throughout. Did the authors check for normality of the data? They need to specify this.

Indeed, we checked normality of the data distributions using the Shapiro-Wilk test. This is stated in the text now.

10) All dendritic images presented in the manuscript appear saturated, which makes it difficult to judge the accuracy of spine counting and the health of dendrite.

We usually adjusted image contrast and brightness so that the spines, especially their necks would be readily visible at the expense of causing some regions in the dendrite to appear saturated. To see what different brightness levels do to the images, we present the images used in the actual figures in a less saturated way, which look very similar (Author response image 2).

**Author response image 2. respfig2:** 

11) The criteria used in counting spines are a bit vague. For example, in Figure 3A, day 1, spine 10 does not seem to be connected to the dendritic shaft; spine 9 seems to be a small bump from the dendritic shaft, whereas similarly-sized bumps (e.g., between spine 18 and 19) are not counted. The methods of counting need to be documented in great and quantitative detail.

We agree with this criticism and have improved the description and transparency of our analysis of spine density, by providing image z-stacks for the reader to scroll through and adding a clarifying paragraph in the Materials and methods section.

We would like to point out that most of the turnover data was analysed independently and blindly (i.e. blind to the order of the imaging session) by the three first authors, which makes us confident about our main finding of high spine turnover.

The revised manuscript contains a table (Figure 3—source data 1) with an overview of the number of spines and dendritic lengths extracted from the turnover data.

In response to the reviewers’ comments, Author response image 3 shows a figure outlining why spines 9 and 10 was counted as indicated and connected to the dendritic shaft. In contrast, the “bump” highlighted by the white arrow head fell below the threshold.

**Author response image 3. respfig3:** 

(New Figure 2—figure supplement 1; new Figure 3—source data 1; new paragraph in Materials and methods.)